# Investigating the Relationship between the Third Places and the Level of Happiness for Seniors in Taiwan

**DOI:** 10.3390/ijerph17041172

**Published:** 2020-02-12

**Authors:** Jui-che Tu, Kang-Chi Lin, Hong-Yi Chen

**Affiliations:** 1Graduate School of Design, National Yunlin University of Science & Technology, Yunlin 64002, Taiwan; tujc@yuntech.edu.tw; 2College of Mechanical and Automotive Engineering, Zhaoqing University, Zhaoqing 516260, China

**Keywords:** happiness, the third place, successful aging, seniors.

## Abstract

Taiwan has actively promoted the concept of "successful aging" in recent years. The Executive Yuan drafted the *White Paper for Aged Society*, which set the primary goal to enable seniors to deal with daily life more independently. Although ‘third places’ enable seniors to live independently, the third places that Taiwanese seniors like are not well understood. Consequently, by investigating third places, this study will investigate the environment of achieving successful aging and happiness among seniors. This study uses the questionnaire survey, and the data of this study were collected from October to November 2018 in Taichung City Central District. A questionnaire survey was conducted in several administrative agencies and participants were selected by random sampling among the over-55-year-old citizens who were already retired. An estimate of 90% confidence limits with 5% marginal error gave us a sample size of 257. This study finally received 200 efficient samples. The women’s top five choices of third places are the traditional market, supermarket, restaurant, daily necessities shop, and coffee shop. The men’s top five choices of third places are the traditional market, supermarket, daily necessities shop, restaurant, and a friend’s house. For seniors familiar with the concept of third places, the more often they go to third places, the higher happiness they achieve. This result investigates the importance of having awareness of third places for seniors. Therefore, we should encourage them to go to third places and engage in social activities frequently to achieve successful aging.

## 1. Introduction

Internationally, the proportion of a society’s population that is comprised of persons aged 65 or older is called the “aging rate”. If a society’s aging rate exceeds 7%, it is an aging society; if the rate surpasses 14%, it is an aged society; and if it exceeds 20%, it is a super-aged society. Taiwan entered the stage of aging society in 1993, and aged society in 2018. It is predicted to become a super-aged society in 2026, which would mean it took just eight years to go from an aged society to a super-aged society. In 2065, there will be about four persons aged 65 or older for every 10 persons, and one person aged 85 in these four persons [1]. With the transformation of population structure, the "seniors" have become the most imperative issue in the world. 

Due to various social changes, such as longer life expectancy, medical advances, and improved health behaviors, the proportion of old age people in the population is increasing. If older adults cannot live independently (depend on the care of others) and are more frequently bedridden, it is not only pain that they suffer, but also places a heavy burden on their family, society, and the country. The government has a greater responsibility for supporting older adults and in prolonging independent living with increased support provided by the social network [2].

Although society nowadays can provide services for the elderly such as nursing care, medical care, housing and community care, as well as welfare institution care, some of the welfare policies for seniors fail to provide sufficient care to all who need it because of some restrictions on their qualifications or their conditions. These seniors who are not properly cared for need to be informed and help by community-based support system in order to have access to social well-being.

Taiwan has actively promoted the concept of “successful aging” since 2007 [3]. The so-called successful aging includes three aspects toward the best mental condition and having a happy old-age life. They include the following: physiological freedom from disease and disability; psychological adaptability, high cognition, and freedom from melancholia; and good family and social relationships [4]. In Taiwan, the proportion of seniors in pension institutions is relatively low, and their intention to stay in old-age institutions is also very low. If seniors cannot live in a familiar place, they will think that they have been abandoned by their families, their happiness will decrease, and the goal of successful aging will become difficult to achieve. Therefore, since 2016, the main goal of the Taiwanese government’s long-term care policy has been to achieve "aging in place" [3]. The idea of Long-term care community-based support system or aging in place is emphasized, not only because of concern about the increasing number of seniors’ considerations but also the cultural considerations.

Currently, most researchers in Taiwan are focusing on investigating the nursing care of disabled seniors, or on dividing the seniors’ needs into senior learning and leisure. In fact, most seniors are afraid of facing the aging process, or of being less needed at home and in the workplace [5]. The third places can make up for the sense of loss in the family and workplace, and provide positive activities and social occasions for seniors. In addition, long-term care institutions and community-based support systems are regarded as a negative term by most seniors in Taiwan. Once the elderly who are accustomed to going to third places begin to lose their abilities and need to use long-term care institutions, the habits obtained in those third places will make them less likely to reject care, so we need to create places that are popular with healthy or sub-healthy seniors. Even places that can make the seniors healthier. Some scholars believed that happiness is the strongest predictor of health, and showed that this is indeed the case [6]. Asking seniors if they are happy is easier to get positive feedback than asking if they are healthy, so this study uses the level of happiness to confirm the importance of the third place. According to the Third Place Theory proposed by sociologist Ray Oldenburg [7], the purpose of this study is to investigate the third places to which seniors nowadays in Taiwan would usually go and determine the importance of third places with respect to the levels of happiness.

## 2. Literature Reviews 

This study includes the relevant theory of successful aging, the definition and characteristics of the third places, and the definition and cognition of happiness. In consequence, the foundation of the design research method is established.

### 2.1. The Third Place

The concept of “third place” was introduced in the American sociologist Oldenburg’s book [7] “The great good place: Cafes, coffee shops, bookstores, bars, hair salons, and other hangouts at the heart of a community”. “The first place” refers to home living space, “the second place” refers to the workplace, and “the third place” refers to informal public gathering places apart from the first and second places.

Oldenburg believes that “the third place” is one of the important elements that composes a good life. However, there was no English word that could express the concept of informal public gathering places at the time, so he proposed the term “the third place” to express the core settings of informal public life, and defined it as a generic term for informal public places apart from work and the home, which have regular, voluntary, and informal gatherings.

Oldenburg considered the third place should have eight basic characteristics: neutral ground, leveler (a leveling place), conversation is the main activity, accessibility and accommodation, the regulars, a low profile, the mood is playful, a home away from home. He listed the following types of possible third places: coffee shops, bars, book stores, gyms, hair salons, movie theaters, art galleries, and other public places in which you can relax. Frumkin et al. associate third places with public health [8,9].

Seniors in Taiwan believe they can still play an important role in the family. However, differences in Taiwan's population ratio and family structure have also changed the seniors’ plans and expectations for family life. Some scholars believe that the focus of the life of the seniors should be able to enrich themselves, not only being limited to their own families [10], and the participation of the seniors through social activities can not only enrich their daily life, but also give them a sense of belonging and happiness [11]. More and more seniors have less of a sense of existence in their first and second places. Thus, the necessity of third places has become higher. The third place is a feasible solution for prompting hanging out and preventing isolation. It brings benefits to the maintenance and promotion of the senior citizens’ physical and mental health [12].

The third place offers stress relief from the everyday demands of both home and work. It also provides a feeling of safety and security by being publicly accessible and promoting open and visible interaction [13].

According to Taiwanese government’s long-term care policy (Long-Term Care 2.0 in Taiwan) in 2017, constructing the comprehensive community care service system creates a comprehensive care system that integrates medical care, LTC(Long-Term Care) services, housing, prevention, and social assistance to allow people with disability to receive the care they need within a 30-minute drive. The smallest service station is Tier C—LTC stations around the blocks must be reachable within 10 minutes on foot, and its main function is to provide respite service in the neighborhood. However, long-term care and community care service system are regarded as a negative term by most seniors in Taiwan, so healthy seniors have very low utilization and high rejection [3]. The third place is a social place where healthy and sub-healthy seniors gather naturally. In addition to the various social and other possibilities inherent to third places (such as shopping, dining, lifelong learning, sports, etc.), we can encourage more seniors to go there and promote health information and long-term care policies in the third place. Exploring the factors that make third places popular can create more popular long-term care facilities. 

### 2.2. Successful Aging

Successful aging has many synonyms such as active aging, productive aging, harmonious aging, healthy aging, vital aging, positive aging, robust aging, optimal aging, and aging well [14,15].

Research on successful aging has pointed out that a happy senior life must fulfill three important factors: physiological, psychological, and social [16,17,18]. The model of successful aging defined by Rowe and Kahn includes three factors: avoidance of disease and disability, maintenance of physical and cognitive function, and engagement in social and productive activities. Rowe and Kahn divided the aging into usual aging and successful aging. General aging refers to non-morbid but high disease risk, and successful aging means that the seniors have the ability to maintain an independent daily life [4]. Some scholars believed that the continuous participation in various social activities can lead the senior citizens towards successful aging [19,20].

Baltes and Baltes used the concepts of variability and plasticity to define successful aging as a process of psychological adaptation, which contains three elements: selection, optimization, and compensation, referred to as the SOC model. Baltes believed that in order to achieve successful aging, the key point is how to choose and adapt to a suitable lifestyle, and then, through learning, training to maintain the optimum state. It can make up for the original motivation that decreases or disappears with age and can live an efficient and functional life after adjustment [21]. For example, choose the third place that is suitable for the needs of seniors, and make use of the activities in the third place to make up for the ever-decreasing sense of need and happiness in families and workplaces, and to help seniors continue to live in the best state of life after adjustment.

George LK proposed Continuity Theory, also called Personality Theory. It mainly emphasizes that senior citizens have consistency in their activities, personalities, and relationships despite their declining physical, mental, and social status, from their adolescence, early adulthood and midlife, and considered that successful aging can only be achieved by maintaining the stability of individuals in activities, personalities and social relationships [22]. Continuity Theory advocates that people should keep engaging in activities in old age, keep their past lifestyle in financial management and social relationships, and maintain their interactions with society [23]. 

### 2.3. Happiness

Happiness is quite a subjective and personal experience [24]. According to the well-being defined by Andrews and Withey, it is a feeling formed by the degree of satisfaction with life, and by the overall evaluation on both positive and negative emotions [25]. Cognition and emotion are regarded as the components of well-being, and they are combined into one inseparable concept and converted to well-being.

Psychological well-being is a well-known component of the Sociology of Aging. There is a huge overlap among life satisfaction, happiness, moral and psychological well-being. Some scholars evaluated it with autonomy, environmental mastery, personal growth, positive relations with others, purpose in life, and self-acceptance [26,27].

Tang HJ conducted her research on psychological happiness satisfaction of seniors lived in southern Taiwan. She divided happiness into four dimensions: take control of life and self-growth, positive relationships with others, autonomy, and self-satisfaction. The study found that if seniors can achieve all the above four dimensions, their overall happiness condition will be good [28]. Neighbors can be of great help to seniors, as they can provide necessary support [29]. Social activities and social participation improve happiness [30]. Neighborhood social networks are important to enhance the health and happiness of seniors, and geographic accessibility is an important factor in these social interactions [31]. Third places provide better geographic accessibility. Based on the above literature, happiness and successful aging are closely related to the third place.

## 3. Materials and Methods

Based on research purpose, the communication approach of this study used self-administered survey, and the questionnaire including “personal basic data “, “third places” and “happiness”. In order to increase the validity of the questionnaire, face-to-face interview was conducted. Thirty seniors and caregivers were invited and interviewed. An initial survey instrument was tested to verify if the survey questions were clear in meaning and wording. After receiving feedback from respondents, appropriate adjustments were made to fit seniors living in Taiwan. 

The data of this study were collected from October to November 2018. This study used Taichung City Central District, the fastest aging population in Taiwan, as the research area. The Taichung Cultural City Midtown Project provides activities for local people in the Central District, while promoting community-friendly relations, and planning many open multi-functional spaces for rest and use. The area has a population of 18,557, and there were 5458 (29.4%) people aged over 55 years in October 2018 (Source: Central District Office). A questionnaire study was conducted in several administrative agencies, and selected by random sampling the over-55-year-old citizens who were already retired collectors through oral inquiry. An estimate of 90% confidence limit with 5% marginal error gave us a sample size of 257. This study finally received 200 efficient samples. The response rate was 77.28 percent.

The questionnaire of this study was divided into three parts. The first part requested personal basic data including six terms: gender, age, education, marital status, financial status, and living situation. Two items among them—marital status and living situation—were included to understand the current conditions in the first places of senior citizens.

The second part was a survey of third places. It included whether the participants understood the concept of third places, their third places, their social places, and the frequency with which they visited those places.

The third part was a questionnaire of happiness, and we obtained the happiness average value in this part. The study used the Oxford Happiness Questionnaire (OHQ), as we could obtain the results of questionnaire rapidly, so the interaction with older participants was better. The OHQ is derived from the Oxford Happiness Inventory (OHI). The OHI was designed by Argyle and used to survey English people’s happiness scale. It pointed out that English people’s happiness consists of seven major sources: optimism, social commitment, positive emotion, sense of control, physical health, self-satisfaction, and psychological vigilance. OHI comprises 29 items, each involving the selection of one of four options that are different for each item.

The OHQ includes similar items to those of the OHI, each presented as a single statement which can be endorsed on a uniform 6-point Likert scale. The revised instrument is compact, easy to administer and allows endorsements over an extended range. When tested against the OHI, the validity of the OHQ was satisfactory and the associations between the scales and a battery of personality variables known to be associated with well-being were stronger for the OHQ than for the OHI. The scores range from one to six, and the average score can be used to determine the level of happiness. The higher the score, the higher the level of happiness [32,33].

## 4. Discussion

### 4.1. Characteristics of the Study Respondents 

In efficient samples, Table 1 shows that the total number of female participants was 121 (60.5%), and the total number of male participants was 79 (39.5%). In age distribution, the largest number of participants were aged 60–64 years (77 people), with 38.5% of the total efficient samples. This was followed by the age groups 55–59 years (46 people, 23%), 65–69 years (39 people, 19.5%), 70–74 years (17 people, 8.5%), 75–79 years (11 people, 5.5%), and over 80 years (10 people, 5%). In terms of education level distribution, the largest number of participants were “high school” (83 people, 41.5%). Most participants’ marital status was “married” (153 people, 76.5%). In financial status distribution, the largest number of participants was seen at “enough” (166 people, 83%). In terms of living situation distribution, the largest number of participants was seen at “live with two generations (at least) of family” (107 people, 53.5%). This was followed by “live only with spouse” (63 people, 31.5%), “live alone” (17 people, 8.5%), “live with brother, sister, or friend” (9 people, 4.5%), and “live in child’s home” (4 people, 2%)

### 4.2. Comparison of Happiness Levels between Awareness and Unawareness of Third Places

When pretesting the questionnaire, seniors who refused to go to social places had a low willingness to answer the questionnaire. In the formal questionnaire survey, the subjects were changed to elderly people who would go to social places and asked if they knew the third place theory. This study used a t-test to examine the difference between the awareness of third places and happiness, where the independent variables was awareness of third places and the dependent variable was happiness levels. The awareness of third places refers to the seniors who know the definition of the third place and have their own favorite third places. The number of the participants having the awareness of third places is 77 (38.5%), while those not aware is 123 (61.5%). Those in the former group had a mean of happiness of 4.05/6.0, while for the latter it was 3.83/6.0 (see Table 2). Results show that the participants with awareness of third places have more happiness than those that do not. 

### 4.3. Compare the Happiness Levels among Different Participants’ Frequency

For more detailed analysis, we check the happiness levels among different participants’ frequency. This study used one-way ANOVA to compare the mean of happiness between groups of participants considering the awareness of third place; the independent variable was frequency and the dependent variable was happiness level. Table 3 reveals the awareness of third places and frequency with the mean of happiness. The participants with an awareness of third places who go to social places at least once a day have a mean happiness of 4.27/6.0. This was followed by “two or three times a week” (4.00/6.0), “once a week” (3.93/6.0), “other frequency” (3.72/6.0), and “once or twice a month” (3.65/6.0). 

In comparison, the participants without an awareness of third places who go to social places in other frequency had a happiness value of 3.93/6.0. This was followed by “two or three times a week” (3.91/6.0), “once a week” and “once or twice a month” (3.77/6.0), and “at least once a day” (3.6/6.07). It was concluded that there is no significant difference in mean awareness of third places and frequency of visits with the mean of happiness. Although these differences were not significant, it is still worthy of further study. For broader perspective it would be possible to assume, that lower happiness may result in lower motivation to go out to third places.

### 4.4. Third Places of Seniors in Taiwan

The data for social places of seniors in Taiwan are shown in Table 4. They reveal that the women’s top five choices of third places are traditional market (22.5%), supermarket (16.2%), restaurant (9.7%), daily necessities shop (8.4%), and coffee shop (7.6%). The men’s top five choices of third places are traditional market (15.3%), supermarket (13.9%), daily necessities shop (10.2%), restaurant (9.7%), and friend’s house (9.7%). Older women might benefit more than men in terms of happiness from experiencing positive social exchanges [34]; therefore, the third place behavior of gender differences needs to be further studied.

### 4.5. Happiness and Living Situation 

This study used ANOVA to determine the difference between the mean of happiness and living situation. Table 5 shows the happiness value of seniors with living situation. The average value of happiness of the participants living alone is 4.08, followed by “live with friend, brother, or sister” (4.01), “live with each child by turns” (3.95), “live with spouse” (3.92), and “live with two generations (at least) of family" (3.88). It was concluded that there is no significant difference in mean of happiness due to living situation. However, senior citizens that live with more family members have less happiness, as shown in Figure 1. This is an unusual result and inconsistent with the traditional Chinese cultural “bigger families are happier" belief in Taiwan.

There may be some possible limitations in this study. This study is based on study time and conditions, so the research object was seniors living in the metropolitan area of Taiwan. The factors affecting social willingness and ability to go out are complex and diverse. This study only studies the gender, age, education level, marital status, financial status, and living situation of seniors, and does not discuss other factors. This study adopts the questionnaire survey method for investigation. Therefore, it is related to the respondents' understanding of the meaning of the question and the self-evaluation, which may cause errors in measurement. Therefore, the inference of the results of this study is limited to literate seniors living in the metropolitan area of Taiwan.

## 5. Conclusions

The concept of the third place is not popular in Taiwan, and there are few studies. There are many studies on the happiness of seniors in various aspects, but there are few researches on the happiness of the living and social places of retired people. Based on the above literature review, aging in place is a very important social issue and government policy, and lifestyle in Taiwan is very encouraging for seniors to go out of their home into the community. The third place satisfies such living needs. Once the third place that seniors like is determined and a survey of happiness is used, various possibilities for the third place can be obtained.

According to the results of this study, some conclusions can be made as follows:

1. Results show that senior citizens with awareness of third places have more happiness than those that do not. Whether or not senior citizens are unaware of third places, there is no significant correlation between their frequency of going to third places and happiness. Therefore, we should encourage them to go to third places and engage in social activities frequently to achieve successful aging.

2. The traditional market and supermarket are the favorite third places for senior citizens in Taiwan. The traditional market especially occupies a very high proportion in the choices of female seniors, while there is no item of significantly high proportions in the choices of male seniors. In addition, the proportion of those choosing “restaurant” is much higher in male senior citizens than in female senior citizens. Moreover, there is only one difference in the top five choices of men and women: for men, this is the inclusion of “friend’s house”, and for women “restaurant”. These two results are worth researching further. 

The Taiwan government uses most of senior welfare budget for community centers, senior citizen adult education institutions, and day care centers. In this study, the latter two types of place (senior citizen adult education institution and day care center) are low-ranking among the favorite social places for senior citizens. This situation should be reviewed and improved. In order to encourage more non-governmental organizations to participate in the aging industry, and even more people can participate in local creation, the results of this study are expected to give more choices to operators and users of third places, caregivers of senior citizens and senior citizens, and building an age-friendly environment.

In addition, the research object of this study is limited to healthy seniors living in metropolitan areas. The follow-up research objects should be seniors living in rural areas and remote areas, and investigate the needs of seniors with reduced mobility. Further study should focus on the attraction of different third places for seniors, the different choices between seniors with different lifestyles, and the correlation between happiness and various third places. The senior citizens in Taiwan have very low efficiency and willingness to fill out the questionnaires. In this context, an observation method and interview method are suggested to use for deeper research in the senior citizens’ preferences and behaviors related to third places.

## Figures and Tables

**Figure 1 ijerph-17-01172-f001:**
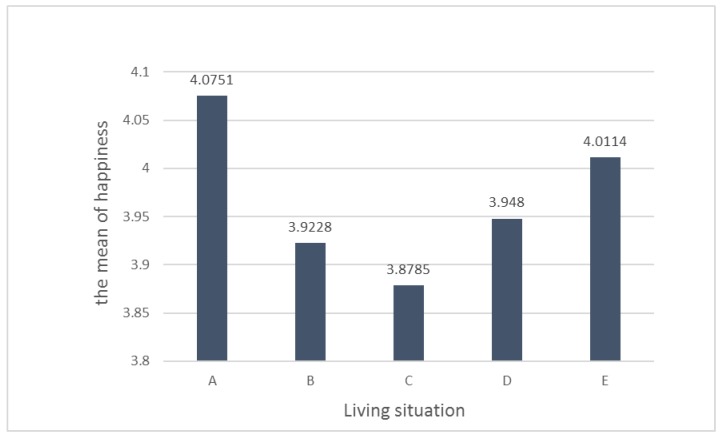
Bar chart for living situation distribution and the mean of happiness.

**Table 1 ijerph-17-01172-t001:** The characteristics of the study respondents (*n* = 200).

Characteristics	Category	Number	Percent (%)
**Gender**	Male	79	39.5
	Female	121	60.5
	Total	200	100
**Age**	55–59	46	23
	60–64	77	38.5
	65–69	39	19.5
	70–74	17	8.5
	75–79	11	5.5
	>80	10	5
**Education level**	Illiterate	2	1
	Elementary school	8	4
	Junior high school	16	8
	High school or vocational high school	83	41.5
	University or junior college	78	39
	Master’s degree	10	5
	Ph.D.	3	1.5
**Marital status**	Married	153	76.5
	Divorced or separated	9	4.5
	Widowed	26	13
	Single	12	6
**Financial status**	Rich	18	9
	Enough	166	83
	Slightly difficult	14	7
	Quite difficult	2	1
**Living situation**	Live alone	17	8.5
	Live only with spouse	63	31.5
	Live with two generations (at least) family	107	53.5
	Live with each child by turns	4	2
	Live with brother, sister, or friend	9	4.5

**Table 2 ijerph-17-01172-t002:** Awareness of third places and the mean of happiness (based on 6.0 scale).

Category	Awareness of Third Places		Unawareness of Third Places		t-Test
	M	SD	M	SD	
**Happiness**	4.0525	0.6929	3.8315	0.5515	2.494

*p* = 0.013, *N* = 200

**Table 3 ijerph-17-01172-t003:** ANOVA summary table for awareness of third places and frequency of visits with the mean of happiness.

Awareness of Third Places	Frequency	*N*	Mean	SD	SE	95% Confidence Interval for Mean		MIN	MAX
						Lower Bound	Upper Bound		
	At least once a day	19	4.2741	0.59918	0.13746	3.9853	4.5629	3.07	5.17
	Two or three times a week	42	4.0083	0.67532	0.1042	3.7979	4.2188	2.52	5.48
**Yes**	Once a week	14	3.9359	0.87104	0.23279	3.433	4.4388	2.72	5.59
	Once or Twice a month	1	3.655	.	.	.	.	3.66	3.66
	Other frequency	1	3.724	.	.	.	.	3.72	3.72
	Total	77	4.0525	0.69297	0.07897	3.8952	4.2097	2.52	5.59
	At least once a day	24	3.6666	0.60776	0.12406	3.41	3.9233	2.66	4.79
	Two or three times a week	58	3.9149	0.54432	0.07147	3.7718	4.0581	2.72	5.72
**No**	Once a week	29	3.774	0.48459	0.08999	3.5897	3.9584	2.66	4.69
	Once or Twice a month	11	3.8935	0.62861	0.18953	3.4712	4.3159	3.17	5.35
	Other frequency	1	3.931	.	.	.	.	3.93	3.93
	Total	123	3.8315	0.55156	0.04973	3.733	3.9299	2.66	5.72
**ANOVA**									
**Awareness of third places**		**Sum of Squares**	**df**		**Mean Square**		**F**	**Sig.**	
**YES**	Between Groups	1.471	4		0.368		0.756	0.557	
	Within Groups	35.024	72		0.486				
	Total	36.495	76						
**No**	Between Groups	1.204	4		0.301		0.989	0.416	
	Within Groups	35.911	118		0.304				
	Total	37.115	122						

**Table 4 ijerph-17-01172-t004:** The third places of seniors in Taiwan.

The Third Place	Number	Male	Ranking	Number	Female	Ranking
**Bookstore**	12	5.60%	9	21	5.50%	8
**Restaurant**	21	9.70%	4	37	9.70%	3
**Coffee shop**	13	6.00%	8	29	7.60%	5
**Traditional market**	33	15.30%	1	86	22.50%	1
**Supermarket**	30	13.90%	2	62	16.20%	2
**Daily necessities shop**	22	10.20%	3	32	8.40%	4
**Library**	15	6.90%	6	14	3.70%	10
**Senior citizen adult education program**	3	1.40%	14	8	2.10%	13
**Friend’s house**	21	9.70%	4	25	6.50%	6
**Community center**	15	7%	6	25	6.50%	6
**Place of worship**	6	2.80%	12	12	3.10%	11
**Medical institution**	11	5.10%	10	9	2.30%	12
**Day Care Center**	0	0.00%	15	1	0.30%	15
**Scenic area**	9	4.20%	11	16	4.20%	8
**Park**	5	2.30%	13	6	1.60%	14
**Total**	216	100.00%		383	100.00%	

**Table 5 ijerph-17-01172-t005:** ANOVA summary table for living situation distribution and the mean of happiness.

Living Situation	N	Mean	SD	SE	95% Confidence Interval for Mean	MIN	MAX
Lower Bound	Upper Bound
A	Live alone	17	4.0751	0.79238	0.19218	3.6677	4.4825	2.72	5.72
B	Live only with spouse	63	3.9228	0.58326	0.07348	3.7759	4.0697	2.66	5.59
C	Live with two generations (at least) of family	107	3.8785	0.61994	0.05993	3.7597	3.9973	2.52	5.48
D	Live with each child by turns	4	3.948	0.55637	0.27819	3.0627	4.8333	3.45	4.59
E	Other frequency	9	4.0114	0.5532	0.1844	3.5862	4.4367	3.17	5
	Total	200	3.9166	0.61767	0.04368	3.8304	4.0027	2.52	5.72
**ANOVA**					
**Living situation**	**Sum of Squares**	**df**	**Mean Square**	**F**	**Sig.**
Between Groups	0.669	4	0.167	0.434	0.784
Within Groups	75.253	195	0.386		
Total	75.923	199

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
