# Peer review of "Investigating the Relationship between the Third Places and the Level of Happiness for Seniors in Taiwan"

_ijerph, 2020, doi:10.3390/ijerph17041172_

Round 1
Reviewer 1 Report
Dear Authors,
In comparison to the previous version, the manuscript has been added extra content, which increases its' quality. However, main issues I had raised previously, remained unresolved.
The basic conclusion coming from the study regards differences in the level of happiness between seniors who are aware and unaware of the concept of "third places". At the same time, both groups compared in the study, did attend third places, which was shown in the frequencies provided in the manuscript. It is a major conceptual issue to me, as I lack understanding of the meaningfulness of this result - it is still unclear from the manuscript whether it should be utilization of third places or ability to define the concept that shall be responsible for well-being of seniors. First step to help with this doubt would be elaborating on the way of measuring understanding of third places, maybe it's a matter of lack of clarity about measurement. Another option would be elaborating discussion section and adding information about why it's rather the understanding of the concept than actual attending third places that seems to be helpful, would any of the theories or results presented in the Introduction be helpful? What practical implication would this result have? How would it translate into practice? I cannot see it from the description provided.
Also, it is hypothesized that greater frequency of attending third places could result in higher overall happiness of participants. Although these differences were not significant, they are quite thoroughly discussed in the text. For broader perspective it would be possible to assume, that lower happiness may result in lower motivation to go out to third places.
Two conclusions are not supported in the text. There are no results at all presented in the manuscript for conclusion number 1, and the differences in happiness related to frequency of attending third places among participants aware of the concept are not significant (according to data presented in Table 3). Conclusion 2 refers to correlations, while no correlational analyses were presented in the manuscript.
Also, there is incongruency between Table 5 and Figure 1, which present the same data – corresponding means are not the same, which is very confusing.
The results that have been presented have got limitations, not discussed in the manuscript. One example would be specificity of the study group – regarding age and marital status of participants, another – violating assumptions for the statistical tests applied. Also, the response rate in the study hasn’t been added.
I believe that further elaboration on the conceptual level, as well as with regard to obtained results would be necessary.
Reviewer 2 Report
Thank you for your corrections to the paper.
Reviewer 3 Report
This revised manuscript is improved, but still missing some important components. The revisions provide more depth to the manuscript and better orient the reader, but the the findings discussion is lacking reference to relevant literature. How are the findings of this study similar or different from other studies? While third places may not be a term used in other cultures, there are studies on the importance of community engagement, social engagement, and so on that positively impacts health, wellbeing, successful aging, and happiness. The conclusion is great, but I would recommend adding a section before it that discusses the findings in more context to the current literature. I would also recommend having an editor check the language and sentence structure for clarity.
Round 2
Reviewer 1 Report
Dear Authors,
I read the revised manuscript with pleasure and I was glad to see the changes. The subject you discuss is very important and the results support the idea of aging in place.
I do have two small comments, which don't belittle value of the text. First one, regards the place where the limitations are discussed - typically, such element would be included at the end of the discussion section, rather than methods.
Second comment is related to the quality of English. Despite the fact I don't feel qualified to make assessments of language mistakes, as I'm not a native speaker, I believe some additional language editing would be necessary prior to publishing.
Reviewer 3 Report
This manuscript has been revised in response to the reviewer's comments, and the paper is improved, but needs continued editing with the new additions to the text and ensuring the writing is concise. This is a great study and will be an important contribution to the literature, but needs a little more editing.

Author Response
Please see the attachment.

This manuscript is a resubmission of an earlier submission. The following is a list of the peer review reports and author responses from that submission.
Round 1
Reviewer 1 Report
Dear Authors,
It was really interesting to read your article regarding well-being of older adults, thank you for this opportunity. The subject of your study seems to be very promising, which is rather clearly pointed in the Introduction. However, overall merit of the research remains unclear to me.
First of all, it is not sufficiently described what research questions or hypotheses were investigated. Also, neither of the variables is defined, which makes it impossible to understand presented results. The variable referred to throughout Discussion section is „ awareness of third places” and „social places”, but they were not operationalized in the manuscript. Subsequently, meaningfulness of the presented results is difficult to establish. Suitability of applied methods may raise doubts, as sample sizes in some of compared subgroups are low (N<20), which violates assumptions for using analysis of variance; no information regarding normality of distribution was reported. Some of the conclusions presented in the manuscript pertain to data, which was not included in analyses (conclusion 1).
Problems presented above are of great importance, because they impact significance and meaningfulness of the study.
The manuscript’s structure need to be edited thoroughly: Results section needs to be added, Discussion section should be elaborated and reorganized (currently, it consists of description of results), conclusions should regard only analyses presented in the manuscript. Methods section should be further developed – OHQ is described mainly in regard to other tool (OHI), no psychometric properties of either one is provided. From the description of the procedure it is not clear how participants were approached, number of refusals hasn’t been indicated.
Presentation of the results need to be improved – there is no need to repeat all information included in the tables in the text (as in case of Table 1). Tables 3 and 5 include unnecessary details.
Overall, I believe, that the manuscript in a current form wouldn't be suitable for publication.
Reviewer 2 Report
highly relevant to the Taiwanese sample describe here, given their rapidly aging population.
-promotion of social health and wellbeing
-investigate connection between third place and happiness
-would avoid the word “prove”. Could use determine/examine/investigate/seek evidence for
2 line 68. Repetition, great and good-I’m not sure what comparison you are showing in Table 3. Can you please, in the body of the text, compare happiness levels between people with and without awareness of third places? What is your within subjects variable?
-organize Table 4 by frequency
-define your independent and dependent variables for your analyses
-Figure on Page 8. Living condition is a categorical variable. Best to display these data in a table or in a bar chart
-Analysis 4.2- why do you call this a correlation?
-p. 8, line 224- what do you mean by correlation?
Reviewer 3 Report
This manuscript has the potential to make significant contributions to the field of gerontology and aging-friendly city movements. The findings of this study are significant and provide evidence and support for cities, or countries, to shift their focus on what older adults wish to have, or use, as they age and want to maintain social connections.
This manuscript requires extensive revisions and restructuring sections (i.e., literature review, methods) and further reading and understanding of the literature. The content is all applicable and relevant to the study, but needs some reorganization. I have attached the document with my comments throughout.
